# Soft Contrastive Learning for Visual Localization

**Janine Thoma**[1]      **Danda Pani Paudel**[1]      **Luc Van Gool**[1,2]

[1] Computer Vision Lab        [2] VISICS, ESAT/PSI
ETH Zurich, Switzerland       KU Leuven, Belgium

## Abstract

Localization by image retrieval is inexpensive and scalable due to its simple mapping and matching techniques. The localization accuracy, however, depends on the quality of the underlying image features, often obtained using contrastive learning. Most contrastive learning strategies learn features that distinguish between different classes. In the context of localization, however, there is no natural definition of classes. Therefore, images are artificially separated into positive/negative classes with respect to the chosen anchor images, based on some geometric proximity measure. In this paper, we show why such divisions are problematic for learning localization features. We argue that any artificial division based on a proximity measure is undesirable due to the inherently ambiguous supervision for images near the proximity threshold. To avoid this problem, we propose a novel technique that uses soft positive/negative assignments of images for contrastive learning. Our soft assignment makes a gradual distinction between close and far images in both geometric and feature space. Experiments on four large-scale benchmark datasets demonstrate the superiority of our soft contrastive learning over the state-of-the-art method for retrieval-based visual localization.

## 1 Introduction

A high quality, view aware[1] image often captures sufficient information to uniquely represent a location. Therefore, it is not surprising that we use vision as the primary source of information for localization, navigation, and exploration in our environments. Vision-based localization has become an effective solution for several important applications ranging from robotics to augmented reality. However, the application of such solutions in large scale environments is limited primarily because of the high computational demand. This limitation is usually alleviated using additional sensory systems such as GPS and beacons. In the absence of such sensors—either the device is not equipped with them or the environment denies the usage (e.g. indoors)—image retrieval-based localization is an appealing alternative. The problem of retrieval-based localization equates to matching one or more query images, taken at some unknown location, to a set of geo-tagged reference images.

Traditionally, vision-based localization [1] is tackled either with structure-based methods, such as Structure-from-Motion (SfM) [2, 3, 4, 5, 6, 7] and Simultaneous Localization and Mapping (SLAM) [8, 9, 10, 11, 12], or with retrieval-based approaches [13, 14, 15, 16, 17, 18, 19, 20]. Structure-based methods usually focus on accurate relative pose estimation, while retrieval-based approaches prioritize absolute re-localization. Localization by image retrieval (or simply retrieval-based localization) is inexpensive due to its simple mapping and matching possibilities, which scale well to large spaces [16, 4, 21]. Many structure-based approaches use retrieval for initialization [4]. Recent developments in learning image features for object and place recognition [13, 14, 15, 16] have made image retrieval a viable method for localization.

Almost all current methods for learning image features for localization are based on learning for classification. Discriminative approaches based on contrastive learning in the feature space have recently shown great promise [13, 22] for classification using human supervision. In the absence of such supervision, an effective visual representation can still be learned using the framework of contrastive learning by artificially dividing images into similar and dissimilar categories [23, 24], also known as positives and negatives. When the visual task itself is of contrastive nature, such as face recognition, it is natural to learn features by dividing images into categories. Many notable works [25, 26, 27, 28] in this context, have laid a foundation for learning powerful features when images can be meaningfully divided into *discrete categories*. The state-of-the-art localization features [16, 29, 30, 31] are learned by building upon this foundation, where images are divided into *discrete categories of positives and negatives*, with respect to a chosen anchor image, using some geometric proximity measure to that anchor.

In this paper, we show why dividing images into discrete categories of positive/negative is problematic in the context of learning features for visual localization. Our argument stems from the fact that there is no natural definition of discrete categories of the continuous world. Any artificial division based on some proximity measure immediately becomes undesirable due to the inherently ambiguous supervision for images close to the proximity threshold. To avoid this problem, we propose a novel technique that uses soft positive/negative assignments of images for contrastive learning. We, however, acknowledge that features for place/landmark recognition [16, 14, 29]—which are shown to somewhat generalize for localization—do not necessarily suffer from the same problem, as the landmarks can be discrete. Methods for place recognition primarily aim to distinguish between prominent landmarks—where images do not necessarily have to be geo-tagged[2]. Therefore, place recognition features offer only a vague promise for accurate localization. In this regard, the task of learning localization specific features has received little to no attention in the literature, with the exception of *Thoma et al.* [31]. Although [31] benefits from softly treating the positives, the used positive/negative division directly conflicts with the theoretical argument of our work.

**Contributions.** Our major contributions are threefold. **(i)** We propose a formal theoretical framework, in contrast to the common practice, for learning localization features. **(ii)** Within the proposed framework, we formulate a novel loss function while offering other possibilities to tackle the original multi-objective problem. **(iii)** Using four large-scale datasets, we demonstrate a clear superiority of our method over the state of the art.

## 2  Related Work

**Visual localization features.** Existing works can be broadly divided into: (i) trained and tested on place recognition [32, 18, 33]; (ii) trained on place recognition and tested for localization [16, 29, 30]; (iii) trained and tested for localization [31]. The methods of the first category are designed to learn features to recognize places and, therefore, are tested in the same setup. The second category covers the methods whose formulation targets place recognition (as argued in this paper), but that nevertheless somewhat generalize to the task of visual localization. The third category belongs to methods designed and tested for localization. Our work belongs to the third category. It differs from [31] in the same category, primarily because of the soft vs. hard assignments (see Section 1). Interested readers can refer to [34, 35] for more details on learning localization features.

**Assignments for contrastive learning.** Existing assignment techniques for positives/negatives separation can be broadly divided into: (i) hard assignments [36, 26, 20, 27, 37, 38, 16, 29, 30, 28]; (ii) hybrid assignments [39, 37, 40, 31]; (iii) soft assignments [41]. The hard assignments strictly categorize images into discrete classes (into positives and negatives with respect to the anchor, in our context). Hybrid assignments also use a discrete categorization but allow different treatment for samples within a category. For example, [31] seeks features such that the feature distances are proportional to the corresponding geometric distances for the positive pairs. Soft assignments do not use a discrete categorization of the samples. Our work performs soft assignments yet fundamentally differs from [41] because of the desired property of the sought feature space. In particular, [41] seeks for proportional feature and geometric distances for all pairs (unlike only for positives in [31]). We argue that for geometrically far away samples, it suffices to have large enough feature distances. They do not need to be proportional. Enforcing proportionality for all pairs invites the danger of

memorizing the geometric map. This is the reason why [31] enforces the proportionality only for positives. In our method, only the features close (resp. far) in geometric space but far (resp. close) in feature space are pulled towards (resp. pushed away form) the anchor (see Figure 1). Moreover, our method performs significantly better than [41] in all extensive experiments. Interested readers can refer to [42] for more details and recent developments in contrastive learning.

## 3 Problem Formulation and Background

### 3.1 Preliminaries

Let a set of tuples $\mathcal{T} = \{\mathcal{D}\} = \{(\mathcal{I}, \mathsf{x})\}$ be the given data consisting of pairs of image $\mathcal{I}$ and its geo-location $\mathsf{x}$. We are interested in learning a mapping function that maps images to feature vectors, $\phi_\theta : \mathcal{I} \to \mathsf{f} \in \mathbb{R}^d$, using mapping parameters $\theta$. In the context of this paper, $\phi$ is a convolutional neural network and $\theta$ are the network parameters. For the task of retrieval-based localization, we wish to learn $\theta$ such that the ordering of euclidean distances between features respect the ordering of a geometric proximity measure between the corresponding images[3]. For large scale datasets, measuring the feature distances between all pairs during training is computationally intractable. Therefore, we use the framework of anchor-based learning, which relies on a set of randomly selected anchor features, say $\mathcal{A} = \{\mathsf{a}\} \subset \mathcal{F} = \{\mathsf{f}\}$. In this setup, we wish to address the following problem.

**Problem 3.1** *Find the parameters $\theta$ of the mapping function $\phi_\theta : \mathcal{I} \to \mathsf{f} \in \mathbb{R}^d$, using a given set of tuples $\mathcal{T}$, such that, for any anchor $\mathsf{a} \in \mathcal{A}$ and some geometric proximity measure $\mathbf{d}(.)$,*

$$\mathbf{d}(\mathsf{x}_i, \mathsf{x}_a) \leq \mathbf{d}(\mathsf{x}_j, \mathsf{x}_a), \implies \|\mathsf{f}_i - \mathsf{a}\| \leq \|\mathsf{f}_j - \mathsf{a}\|, \forall \mathcal{D}_i, \mathcal{D}_j \in \mathcal{T}. \tag{1}$$

The choice of euclidean distance in the feature space is made merely to minimize the computation cost of the retrieval process. Note that the condition of (1) suffices to find the geometrically close images using only the feature distances. Before divulging our solution of Problem 3.1, we first revisit a standard approach that has been used so far to learn the mapping parameters $\theta$.

### 3.2 Learning by hard assignments revisited

For every anchor $\mathsf{a} \in \mathcal{A}$, the hard assignment process seeks for two sets of so-called positive and negative examples, say $\mathcal{P}$ and $\mathcal{N}$, respectively. For some arbitrarily chosen geometric proximity thresholds $\tau_1 \leq \tau_2$, positive and negative sets are, $\mathcal{P} = \{\mathsf{f}_i \mid \mathbf{d}(\mathsf{x}_i, \mathsf{x}_a) < \tau_1\} = \{\mathsf{p}\}$ and $\mathcal{N} = \{\mathsf{f}_i \mid \mathbf{d}(\mathsf{x}_i, \mathsf{x}_a) \geq \tau_2\} = \{\mathsf{n}\}$. In this setup, existing approaches address the following problem.

**Problem 3.2** *Find the parameters $\theta$ of the mapping function $\phi_\theta : \mathcal{I} \to \mathsf{f} \in \mathbb{R}^d$, using a given set of tuples $\mathcal{T}$, such that, for any anchor $\mathsf{a} \in \mathcal{A}$, geometric proximity measure $\mathbf{d}(.)$ and a margin $\alpha > 0$,*

$$\max_{\mathsf{p} \in \mathcal{P}} \|\mathsf{p} - \mathsf{a}\| \leq \min_{\mathsf{n} \in \mathcal{N}} \|\mathsf{n} - \mathsf{a}\| - \alpha. \tag{2}$$

Some issues when using the formulation of Problem 3.2 in the context of the retrieval-based localization are: (i) arbitrary choice of $\tau_1$ and $\tau_2$ without treatment of images between the two thresholds; (ii) not implying the desired property of (1); (iii) ambiguous supervision as $\tau_1 \to \tau_2$, which is presented more formally below.

**Proposition 3.3 (Margin ambiguity)** *For a continuous mapping function $\phi_\theta : \mathcal{I} \to \mathsf{f} \in \mathbb{R}^d$ and smooth manifold of features $\mathcal{F}$, when threshold $\tau_1 \to \tau_2$, Problem 3.2 is feasible only if $\alpha \to 0$.*

**Proof** The proof is provided in the supplementary material. ∎

In other words, if there exists a mapping such that feature distances can be ordered according to geometric measures and continuous images are available all over the geometric space, then the margin $\alpha$ must be zero when the two thresholds $\tau_1$ and $\tau_2$ are equal for a feasible solution of Problem 2 to exist. Besides, such a solution does not maintain the desired order.

## 4 Learning by Soft Assignments

Instead of classifying images into positive and negative, we derive two scores for positiveness and negativeness, say $s^+$ and $s^-$, using both feature distances and geometric proximity measures.

**Definition 4.1 (Positiveness and negativeness)** *Let $g^+(y)$ and $g^-(y)$ respectively be decreasing and increasing influence functions with $g^+(y) > 0$, $g^-(y) > 0$ for $y \geq 0$, $g^+(0) > g^-(0)$, and $g^+(y) < g^-(y)$ for sufficiently large $y$. The positiveness and negativeness scores for the image-location tuple $\mathcal{D}_i \in \mathcal{T}$ with respect to an anchor $\mathsf{a} \in \mathcal{A}$ is given by,*

$$s_i^+ = g^+(\mathbf{d}(\mathsf{x}_i, \mathsf{x}_a)) \, \|\mathsf{f}_i - \mathsf{a}\| \quad \text{and} \quad s_i^- = g^-(\mathbf{d}(\mathsf{x}_i, \mathsf{x}_a)) \, \|\mathsf{f}_i - \mathsf{a}\| . \quad (3)$$

Now, we are ready to address Problem 1 with the help of the following proposition.

**Proposition 4.2 (Duality)** *The pareto optimal front of the multi-objective optimization problem, $min_\theta \; [s_1^+, -s_1^-, \ldots, s_i^+, -s_i^-, \ldots]^\top$ passes through the feasible solution set of Problem 1.*

**Proof** The proof is provided in the supplementary material. ∎

Note that Problem 1 with finite cardinality of $\mathcal{T}$ can have a set of feasible solutions, as it lacks an objective function. Any of those solutions is equally valid for the task of visual localization. At this point, finding the exact relationships between feasible and pareto optimal solutions is a topic of future investigation. Henceforth, we use Problem 1 and that of Proposition 4.2 equivalently, with slight abuse of precision. This paper attempts to solve the multi-objective problem instead of addressing the original problem of objective-free constraint optimization. In the following, we will first present an example to clarify Proposition 4.2 in relationship with Problem 2, followed by our attempt at addressing the multi-objective optimization problem. For better clarity of the theory and notations, we present the following example of a special case.

**Example 4.3 (Special case)** *Let the box function be $y = b(x)$ with y=1 for 0<x<1, and y=0 otherwise. For a threshold $\tau$, monotonically increasing/decreasing functions are chosen as, $g^-(y) = b(\tau y)$ and $g^+(y) = 1 - b(\tau y)$. Then, the multi-objective problem $min_\theta \; [\ldots, s_i^+, -s_i^-, \ldots]^\top$ is a dual of Problem 2 with thresholds $\tau_1 = \tau_2$ and the margin $\alpha = 0$.*

Let $\mathsf{u}_a(\mathcal{T}; \theta) = [\ldots, s_i^+ - s_i^-, \ldots]^\top$ be the combined scores of a chosen anchor. In the setup of Example 4.3, methods that address Problem 1 using the well known triplet loss [26] optimize, $min_\theta \; \sum_{\mathsf{a} \in \mathcal{A}} [\mathbf{1}^\top \mathsf{u}_a(\mathcal{T}; \theta) + \alpha]_+$, for the hinge loss $[.]_+$. The setup of [26] involves only one positive and one negative at a time. Similarly, n-pair loss [43] involves one positive and multiple negatives. Notice that for $\alpha = 0$, the triplet loss is optimal when the positiveness score of the positive and the negativeness score of the negative are equal. These scores are kept apart by introducing a margin $\alpha$. In this paper, we show that such a margin is problematic in the context of localization and suggest explicitly maximizing the gap between positiveness and negativeness scores based on our Proposition 4.2. In the following, we will first present our choice of influence functions. Later, we will formulate the loss function, which encourages the gap maximization between the two scores.

### 4.1 Influence functions $g^+$ and $g^-$

Although any choice of strictly increasing and decreasing functions are eligible as influence functions, a careful choice is necessary for the desired outcome in practice. While doing so, two aspects need to be considered: (i) the intersection point of two influence functions represents the threshold $\tau$; (ii) influence functions must be bounded within the interval of the plausible geometric distances. Possible choices are the common linear, log, or sigmoid activation functions. We use the following sigmoid-based influence functions, illustrated in Figure 1, with slope $\gamma$ and offset $\lambda = \tau\gamma$:

$$g^+(y) = \frac{1}{1 + \exp(\lambda - \gamma y)} \quad \text{and} \quad g^-(y) = \frac{1}{1 + \exp(\gamma y - \lambda)}. \quad (4)$$

### 4.2 Loss function for gap maximization

Although any monotone submodular function can be used to combine multiple objectives into a single one; a careful choice is necessary for the desired outcome in practice. Some examples include vector

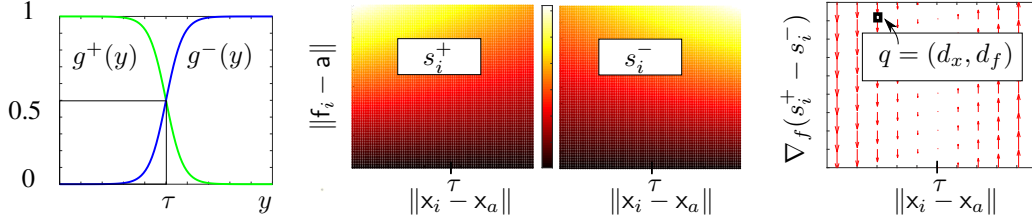

Figure 1: Left to right: influence functions of (4); positiveness and negativeness scores plotted over feature and geometric distances; gradients with respect to feature distance (of the difference of scores as an alternative to (5)) over the space of feature and geometric distances. A point $q = (d_x, d_f)$ represents a pair of geometric and feature distances of an image with respect to some anchor. The gradient at point $q$ shows how the feature f at location x needs to be treated. For example, far away features of close by images are pushed downwards such that the feature distances are minimized.

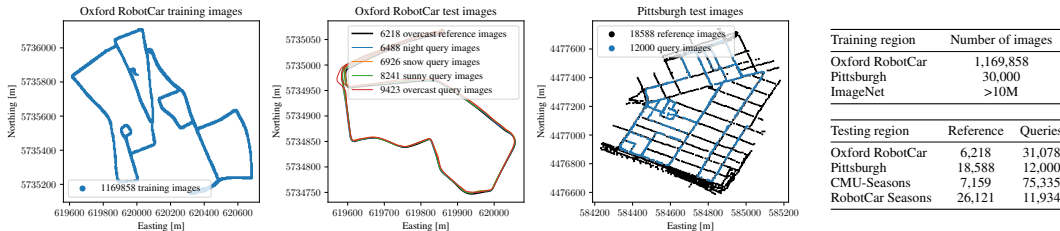

Figure 2: Left to right: training image locations for Oxford RobotCar, test reference and test query image locations for Oxford RobotCar and Pittsburgh, and all training and test set sizes.

norms, weighted norms, or the sum of exponential. Let two vectors $\mathsf{s}_a^+(\mathcal{T}; \theta) = [\ldots, s_i^+, \ldots]^\top$ and $\mathsf{s}_a^-(\mathcal{T}; \theta) = [\ldots, s_i^- \ldots]^\top$ represent positiveness and negativeness scores for a given anchor. Inspired by [28][4], we select a submodular function such that the final optimization problem becomes,

$$\min_\theta \sum_{\mathsf{a} \in \mathcal{A}} \left\{ \log(1 + \mathbf{1}^\top \exp(\eta \mathsf{s}_a^+ - \mu))/\eta + \log(1 + \mathbf{1}^\top \exp(\mu - \nu \mathsf{s}_a^-))/\nu \right\}, \qquad (5)$$

where $\eta, \nu$ represent the slope and $\mu$ represents offset, similarly as in (4). Note that the two terms in (5) are identical, except for the scores and the user controlled slope offset. Intuitively, the first term minimizes the positiveness scores whereas, the second term maximizes the negativeness scores. Recall from (3), both positiveness and negativeness scores are defined for the same image. Note that maximizing the gap between $s_i^+$ and $s_i^-$ pulls feature $\mathsf{f}_i$ closer to a for $\mathbf{d}(\mathsf{x}_i, \mathsf{x}_a) < \tau$. On the other hand, the same objective also pushes $\mathsf{f}_i$ away from a when $\mathbf{d}(\mathsf{x}_i, \mathsf{x}_a) > \tau$. The strength of pulling and pushing depends upon the distance between $\mathbf{d}(\mathsf{x}_i, \mathsf{x}_a)$ and $\tau$, with no action when $\mathbf{d}(\mathsf{x}_i, \mathsf{x}_a) = \tau$. An illustration for this behaviour, with a simple difference maximizing the gap, is shown in Figure 1.

## 5 Experimental Evaluation

**Training datasets.** We conduct experiments with models trained on three different publicly available real world datasets–ImageNet [44], Pittsburgh (Pitts250k) [45], and Oxford RobotCar [46]. The Pittsburgh and Oxford RobotCar models are trained for localization. The ImageNet model, referred to as off-the-shelf, is trained for classification and re-purposed for localization without retraining. For the Pittsburgh and ImageNet models, we use the publicly available checkpoints of [16]. For Oxford RobotCar, we curate our own large-scale training dataset with >1M images, suitable for learning localization features, by carefully removing noisy labels from [46]. Given that the original Oxford RobotCar dataset is somewhat noisy, we filter out bad or atypical locations by removing location outliers and exclude under and overexposed images. The resulting training dataset contains over one million images taken under various conditions, such as night, snow, overcast, sun, clouds, rain, and dusk. The first plot in Figure 2 shows our Oxford RobotCar training image locations. For the sake of reproducibility, we provide a list of all included images in the supplementary material.

**Testing datasets.** We report results on four different test sets: our own Oxford RobotCar [46] test set, Pittsburgh (Pitts250k) [45], RobotCar Seasons (a subset of Oxford RobotCar [46]) and CMU Seasons (a subset of the CMU Visual Localization Dataset [47]). For *Oxford RobotCar*, we select a test region which is geographically disjoint from our training region (see Figure 2). The choice of reference images for our Oxford RobotCar test set is based on two assumptions. Firstly, the reference storage is limited, which is why we only select a limited number of reference images. Secondly, it is justified to choose the easiest condition as a reference, given that anyone designing an image-retrieval based localization system will most likely have control over the initial reference conditions. In that light, we choose an overcast reference sequence. This condition has less glare than sunny sequences and better lighting than night images. In its original form, *Pittsburgh* contains 250k reference images. For faster testing, we ignore all reference images that are more than 100 meters away from the nearest query and remove all images with pitch 2 (i.e. images that mostly show sky). The locations of the remaining images are shown in Figure 2. For reproducibility, we include a list of all Oxford RobotCar and Pittsburgh test images in the supplementary material. For the evaluation on *RobotCar Seasons* and *CMU Seasons* we use the online long-term visual localization benchmark [19]. The table in Figure 2 shows the numbers of reference and query images for the four different test sets.

**Network architecture and training.** We use a VGG-16 [14] network cropped at the last convolutional layer and extend it with a NetVLAD [16] layer as implemented by [48], initializing the network with off-the-shelf ImageNet [44] classification weights, i.e. weights that have not yet been retrained for localization. For dimensionality reduction, we use PCA with whitening. For Pittsburgh, Oxford Seasons, and CMU Seasons, the PCA is calculated using the reference images. For our Oxford RobotCar test set, the PCA is calculated using 5000 images from our Oxford RobotCar training set. All results reported in this paper use a final feature size of 256 (for results with different sizes, please refer to the supplementary material). Please note that this size is much smaller than the 4096 used by [16]. This is the reason why our results obtained with the Pittsburgh checkpoint of [16] on the online benchmark [19] are different from the results of the same method on the leader board.
All models are trained on a single Nvidia GPU. We use the training parameters of [16], reducing the learning rate to 0.000001. Each of our training tuples consists of 1 anchor, 12 close neighbors and 12 further away images. Images are down-scaled such that the largest side has a length of 240px. With the smaller image sizes, it becomes possible to train VGG-16 in its entirety and not only down to conv5 layer as it is done by [16]. Given that our Oxford RobotCar training set contains numerous sequences that run through the same locations, but not every location is visited the same number of times, we redefine the concept of one epoch as having selected one anchor for each one-meter section along the standard driving route. For each anchor image, we sample the close neighbors from within a radius of $r_1$ and with a maximum yaw difference to the anchor of 30 degrees. The further away images are sampled at least $r_2$ distance away. For our method and for log-ratio loss, we set $r_1$ and $r_2$ to 15m. For all other methods we set $r_1$ to 10m and $r_2$ to 25m in accordance with [16, 31, 30]. For our method and log-ratio loss, the close and further away images are merged into one tensor and treated indistinguishably during loss calculation. For the other methods, close neighbors become positives, and further away images become negatives. Similar to [16] we use hard negative mining with a mining cache size of 1000, which is updated every 250 steps. Half of all negatives are mined hard negatives. To assert diversity among negatives, we require that each negative is at least $r_2$ away from all other negatives in the tuple. The training is stopped once the validation performance stops improving. Multi-similarity loss [28] and our method converge within one, log-ratio within two, and all other methods within three epochs.

**Evaluation metric.** We consider an image to be correctly localized if the distance between the top-1 retrieved reference and query location is smaller than a given distance threshold $d$. For any given testing condition, we report the percentage of correctly localized images, i.e. the localization accuracy. The evaluations on CMU Seasons and RobotCar Seasons also use an angle error threshold in addition to the distance threshold. Please note that the long-term visual localization benchmark [19] is designed to evaluate 6DOF structure-based pose estimation methods, while all methods reported in this paper are pure retrieval-based methods. It is possible that even if a method succeeds in finding the geometrically closest reference image, that the pose of that reference is too different from the query image pose and will not be counted as correctly localized. Thus, our evaluations use higher thresholds $d$ and report an upper bound, i.e. the highest possible accuracy achievable by a purely retrieval-based method without a secondary structure-based pose refinement step. Please also note that while structure-based methods often obtain better results at small thresholds, retrieval-based

| | Oxford RobotCar | | | | Pittsburgh | Mean |
|---|---|---|---|---|---|---|
| | Night | Overcast | Snow | Sunny | | |
| Thresholds $d$ [m] | 5.0/10.0/15.0 | 5.0/10.0/15.0 | 5.0/10.0/15.0 | 5.0/10.0/15.0 | 5.0/10.0/15.0 | 5.0/10.0/15.0 |
| Upper bound | 100.0/100.0/100.0 | 92.7/99.3/100.0 | 99.9/100.0/100.0 | 100.0/100.0/100.0 | 57.3/96.8/98.0 | 90.0/99.2/99.6 |
| Off-the-shelf [16] | 4.4/6.5/8.0 | 54.5/69.1/72.1 | 70.2/84.0/86.3 | 76.8/84.7/86.6 | 21.7/44.6/52.2 | 45.5/57.8/61.1 |
| Triplet trained on Pittsburgh [16] | 5.4/7.9/9.3 | 57.8/74.3/79.4 | 76.9/90.5/92.3 | 79.6/87.2/89.3 | **26.0**/52.9/61.5 | 49.1/62.6/66.4 |
| Triplet [16] | 22.7/34.3/36.4 | 60.9/76.8/79.6 | 83.5/95.8/97.2 | 84.6/92.4/93.7 | 24.0/47.9/55.5 | 55.1/69.5/72.5 |
| Quadruplet [27] | 20.2/30.5/33.8 | 59.2/76.0/78.6 | 82.1/94.9/96.5 | 84.8/92.2/94.1 | 23.4/46.6/53.8 | 53.9/68.1/71.3 |
| Lazy triplet [30] | 18.3/24.3/26.5 | 63.9/79.8/82.5 | 83.5/96.3/97.2 | 83.8/91.1/92.9 | 23.2/46.0/53.1 | 54.5/67.5/70.5 |
| Lazy quadruplet [30] | 27.1/40.2/42.3 | 56.6/74.5/77.6 | 84.5/96.1/97.1 | 83.7/93.3/94.8 | 23.5/47.6/55.2 | 55.1/70.4/73.4 |
| Trip. + Huber dist. [31] | 31.6/42.9/45.3 | 56.1/74.0/76.5 | 83.8/95.9/97.2 | 84.2/92.6/93.9 | 21.6/43.6/50.4 | 55.5/69.8/72.6 |
| Log-ratio [41] | 25.0/31.0/33.4 | **69.3**/86.2/88.2 | 86.0/98.1/98.8 | 83.1/94.8/95.5 | 19.6/40.1/47.3 | 56.6/70.0/72.6 |
| Multi-similarity [28] | 30.5/51.3/57.4 | 62.5/85.8/88.5 | 86.6/99.0/**99.7** | 83.4/95.5/97.2 | 24.7/50.8/59.0 | 57.5/76.5/80.4 |
| Ours | **32.9/54.3/60.0** | 67.0/**90.1/93.2** | **87.8/99.1**/99.5 | **89.8/97.1/98.3** | 25.9/**53.1/61.8** | **60.7/78.7/82.6** |

Table 1: Oxford RobotCar and Pittsburgh top-1 localization accuracy. The highest value per condition and threshold for correct localization is marked in bold. All methods except off-the-shelf and triplet trained on Pittsburgh were trained on the Oxford RobotCar dataset.

methods offer easier map building and maintenance. Retrieval-based methods are also used as an initial step in hybrid methods [4].

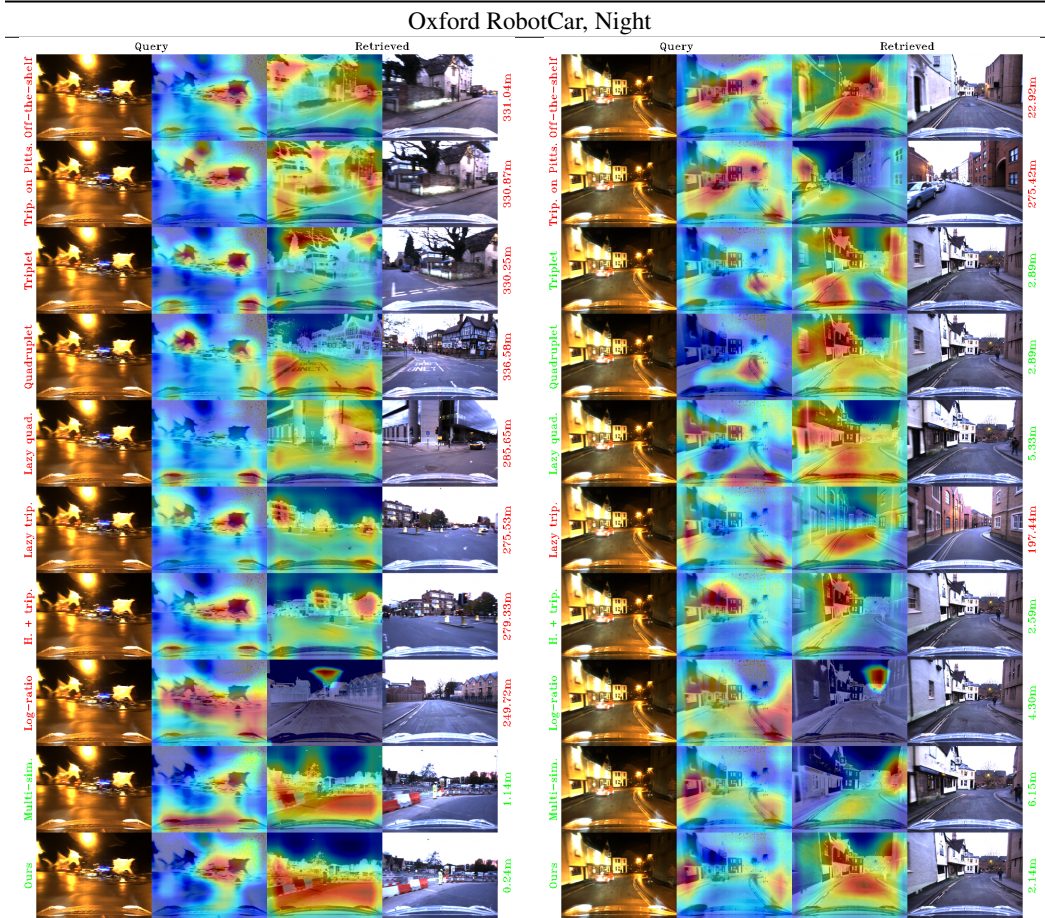

Figure 3: Selected visual results of the oxford robotCar in the night. We show the query image, Grad-CAM on the query image, Grad-CAM on the retrieved image, retrieved image, and the distance between retrieved and query images. Results obtained by our method are shown in the last row.

**Baselines and SoA comparison.** We compare our loss introduced in Section 4 to seven different baselines which we all train on our Oxford Robotcar training set. Our comparison focuses on losses which have previously been studied in the context of localization (triplet loss [16], quadruplet

## Baselines and State of the Art Comparison

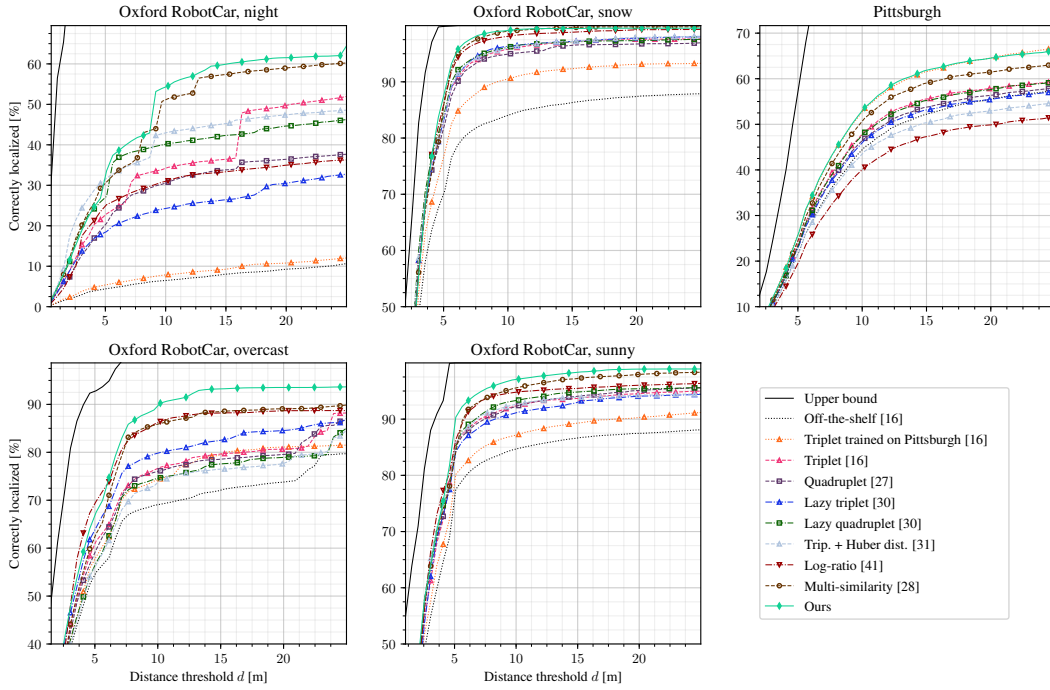

## Influence Function Comparison

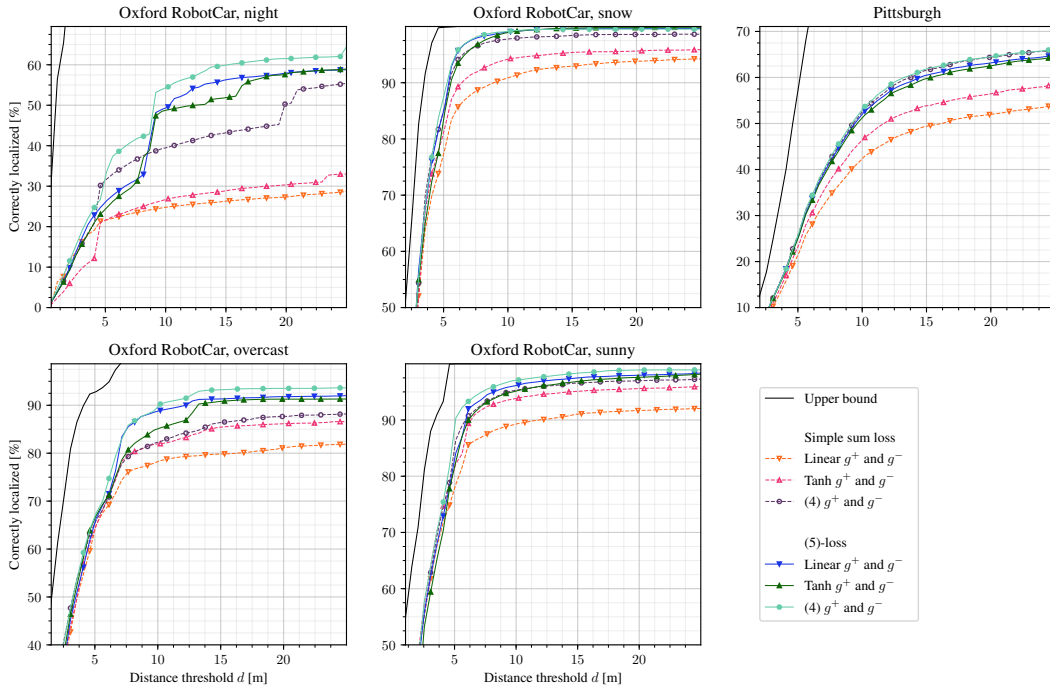

Figure 4: Top-1 localization accuracy on the Oxford RobotCar dataset (four conditions) and Pittsburgh as a function of the distance threshold $d$ for correct retrieval. All methods except off-the-shelf and triplet trained on Pittsburgh were trained on the Oxford RobotCar dataset. The train and test sets do not share any co-visible locations. The top five plots show a comparison between our method and nine other methods. The bottom five plots compare the performance of different influence functions on our method. It also compares our final loss function (5) against a simple sum baseline loss.

Table 2 (RobotCar Seasons, top):

| | Night rain | Overcast winter | Sun | Rain | Snow | Dawn | Dusk | Night | Overcast summer | Mean |
|---|---|---|---|---|---|---|---|---|---|---|
| Thresholds $d$ [m] | .25/.50/5.0 | .25/.50/5.0 | .25/.50/5.0 | .25/.50/5.0 | .25/.50/5.0 | .25/.50/5.0 | .25/.50/5.0 | .25/.50/5.0 | .25/.50/5.0 | .25/.50/5.0 |
| Thresholds [deg] | 2/5/10 | 2/5/10 | 2/5/10 | 2/5/10 | 2/5/10 | 2/5/10 | 2/5/10 | 2/5/10 | 2/5/10 | 2/5/10 |
| Off-the-shelf [16] | 0.2 / 0.2 / 2.5 | 2.1 / 16.4 / 71.3 | 2.4 / 6.5 / 32.0 | 6.9 / 25.4 / 87.9 | 2.9 / 12.7 / 64.2 | 3.5 / 13.0 / 54.9 | 5.6 / 21.3 / 81.0 | 0.0 / 0.5 / 4.1 | 3.2 / 18.6 / 72.4 | 3.0 / 12.7 / 52.3 |
| Triplet trained on Pittsburgh [16] | 0.0 / 0.5 / 3.9 | 2.1 / 16.7 / 73.3 | 2.2 / 7.0 / 40.4 | 7.4 / 27.8 / 90.3 | 5.9 / 17.0 / 72.2 | 3.7 / 13.7 / 56.5 | 7.4 / 24.1 / 80.5 | 0.0 / 0.2 / 5.5 | 3.2 / 19.0 / 74.9 | 3.5 / 14.0 / 55.3 |
| Triplet [16] | 0.2 / 2.0 / 18.2 | 2.8 / 21.0 / 81.0 | 3.9 / 12.2 / 63.3 | 7.4 / 30.4 / 93.3 | 5.7 / 18.8 / 85.1 | 4.8 / 19.3 / 75.8 | 7.1 / 27.4 / 88.8 | 0.5 / 1.6 / 13.5 | 4.3 / 20.5 / 78.6 | 4.1 / 17.0 / 66.4 |
| Quadruplet [27] | 0.5 / 2.7 / 21.1 | 2.3 / 20.8 / 81.3 | 2.2 / 9.3 / 62.0 | 7.6 / 29.2 / 94.8 | 5.1 / 18.6 / 83.0 | 4.6 / 19.3 / 77.8 | 5.8 / 22.8 / 90.6 | 0.2 / 3.0 / 14.6 | 4.8 / 20.3 / 79.3 | 3.7 / 16.2 / 67.2 |
| Lazy triplet [30] | 0.7 / 3.6 / 23.0 | 2.3 / 19.5 / 82.3 | 2.2 / 10.4 / 59.3 | 8.1 / 28.3 / 94.8 | 5.7 / 18.2 / 85.1 | 4.6 / 16.4 / 73.5 | 6.6 / 25.4 / 88.3 | 0.0 / 0.9 / 9.1 | 3.7 / 18.1 / 72.4 | 3.8 / 15.6 / 65.3 |
| Lazy quadruplet [30] | 0.5 / 1.8 / 17.5 | 2.1 / 16.9 / 82.6 | 3.0 / 10.7 / 68.0 | 6.7 / 28.3 / 93.6 | 5.3 / 17.2 / 81.4 | 6.0 / 18.8 / 75.4 | 7.1 / 24.4 / 87.1 | 0.0 / 0.7 / 9.1 | 4.5 / 19.2 / 78.6 | 3.9 / 15.3 / 65.9 |
| Trip. + Huber dist. [31] | 0.0 / 2.0 / 14.3 | 2.1 / 19.5 / 74.1 | 2.2 / 10.4 / 59.3 | 9.3 / 31.8 / 93.8 | 4.3 / 17.4 / 75.7 | 5.8 / 19.3 / 73.3 | 7.1 / 24.6 / 82.0 | 0.0 / 0.9 / 13.2 | 3.9 / 17.5 / 74.1 | 3.9 / 15.9 / 62.2 |
| Log-ratio [41] | 0.2 / 1.4 / 14.8 | 2.3 / 20.3 / 82.1 | 4.1 / 13.3 / 76.7 | 6.4 / 28.7 / 95.7 | 6.1 / 21.1 / 84.3 | 6.0 / 16.6 / 71.8 | 4.6 / 23.6 / 87.6 | 0.0 / 0.2 / 2.3 | 2.8 / 19.2 / 79.3 | 3.6 / 16.0 / 66.1 |
| Multi-similarity [28] | 0.7 / 2.5 / 20.9 | 1.0 / 16.2 / 79.2 | 2.4 / 10.9 / 77.0 | 6.9 / 26.4 / 94.1 | 4.1 / 18.2 / 82.8 | 4.6 / 16.8 / 76.2 | 4.8 / 23.9 / 90.4 | 0.0 / 0.7 / 8.2 | 3.2 / 15.6 / 81.0 | 3.1 / 14.6 / 67.8 |
| Ours | 0.2 / 1.8 / 15.9 | 2.1 / 19.7 / 79.7 | 3.5 / 14.1 / 80.0 | 8.3 / 27.3 / 94.5 | 5.3 / 21.7 / 87.7 | 4.6 / 17.4 / 73.3 | 5.6 / 22.6 / 91.6 | 0.0 / 1.4 / 8.4 | 4.1 / 19.0 / 89.2 | 3.7 / 16.1 / 68.9 |

Table 2 (CMU Seasons, bottom):

| | Urban | Suburban | Park | Overcast | Sunny | Foliage | Mixed foliage | No foliage | Low sun | Cloudy | Snow | Mean |
|---|---|---|---|---|---|---|---|---|---|---|---|---|
| Thresholds $d$ [m] | .25/.50/5.0 | .25/.50/5.0 | .25/.50/5.0 | .25/.50/5.0 | .25/.50/5.0 | .25/.50/5.0 | .25/.50/5.0 | .25/.50/5.0 | .25/.50/5.0 | .25/.50/5.0 | .25/.50/5.0 | .25/.50/5.0 |
| Thresholds [deg] | 2/5/10 | 2/5/10 | 2/5/10 | 2/5/10 | 2/5/10 | 2/5/10 | 2/5/10 | 2/5/10 | 2/5/10 | 2/5/10 | 2/5/10 | 2/5/10 |
| Off-the-shelf [16] | 7.8 / 18.5 / 54.9 | 2.9 / 8.5 / 36.2 | 2.6 / 6.8 / 32.0 | 5.1 / 12.7 / 45.5 | 4.8 / 11.7 / 42.7 | 4.8 / 11.9 / 43.6 | 4.5 / 11.0 / 37.4 | 5.3 / 14.0 / 48.8 | 3.9 / 10.2 / 36.9 | 6.4 / 15.4 / 49.9 | 3.8 / 9.4 / 34.6 | 4.7 / 11.8 / 42.0 |
| Triplet trained on Pittsburgh [16] | 9.1 / 21.5 / 64.3 | 3.2 / 9.5 / 45.0 | 2.5 / 6.9 / 32.7 | 5.6 / 14.1 / 51.8 | 5.1 / 12.5 / 47.1 | 5.2 / 12.9 / 48.6 | 5.2 / 12.7 / 43.7 | 6.3 / 16.0 / 55.4 | 4.6 / 12.2 / 43.4 | 7.0 / 16.7 / 56.0 | 4.5 / 11.3 / 42.2 | 5.3 / 13.3 / 48.2 |
| Triplet [16] | 9.4 / 22.6 / 71.2 | 3.9 / 11.8 / 60.1 | 3.2 / 9.2 / 45.2 | 5.8 / 15.0 / 59.9 | 4.9 / 12.6 / 52.3 | 4.9 / 12.9 / 53.8 | 6.3 / 15.6 / 58.6 | 7.6 / 20.0 / 71.1 | 5.9 / 15.7 / 59.6 | 7.5 / 18.4 / 66.3 | 6.0 / 16.4 / 60.3 | 6.0 / 15.5 / 59.9 |
| Quadruplet [27] | 10.7 / 25.2 / 73.3 | 4.4 / 13.0 / 61.4 | 3.9 / 10.8 / 47.9 | 6.6 / 16.6 / 61.0 | 5.6 / 14.0 / 53.2 | 5.7 / 14.4 / 54.7 | 7.3 / 18.0 / 61.6 | 8.5 / 22.2 / 74.8 | 6.7 / 17.8 / 62.9 | 9.0 / 21.4 / 69.2 | 7.5 / 19.6 / 65.7 | 6.9 / 17.5 / 62.3 |
| Lazy triplet [30] | 9.9 / 23.5 / 69.8 | 4.1 / 11.9 / 58.2 | 3.5 / 10.1 / 42.0 | 5.9 / 14.8 / 54.6 | 5.1 / 12.9 / 47.2 | 5.1 / 12.9 / 48.4 | 6.6 / 16.6 / 57.1 | 8.3 / 22.2 / 75.0 | 6.3 / 16.9 / 60.0 | 8.4 / 20.1 / 65.8 | 7.3 / 19.6 / 66.0 | 6.4 / 16.5 / 58.6 |
| Lazy quadruplet [30] | 11.4 / 26.9 / 72.7 | 4.9 / 13.9 / 64.1 | 3.7 / 10.7 / 44.1 | 7.1 / 17.2 / 59.3 | 5.8 / 14.5 / 49.9 | 5.9 / 14.8 / 51.5 | 7.6 / 18.8 / 60.6 | 9.3 / 24.5 / 77.5 | 7.2 / 19.1 / 63.7 | 9.2 / 22.2 / 67.0 | 8.1 / 21.2 / 68.8 | 7.3 / 18.5 / 61.7 |
| Trip. + Huber dist. [31] | 9.5 / 22.9 / 69.0 | 4.4 / 12.4 / 57.3 | 3.0 / 8.4 / 39.6 | 6.1 / 15.1 / 55.8 | 5.3 / 13.4 / 51.0 | 5.3 / 13.5 / 51.8 | 6.1 / 15.1 / 53.6 | 7.4 / 19.0 / 65.6 | 5.7 / 15.0 / 54.3 | 7.5 / 18.1 / 62.1 | 6.1 / 15.3 / 54.9 | 6.0 / 15.3 / 55.9 |
| Log-ratio [41] | 10.5 / 24.9 / 71.4 | 4.6 / 13.4 / 57.4 | 3.5 / 10.2 / 42.8 | 6.1 / 15.4 / 54.2 | 5.2 / 12.9 / 47.4 | 5.1 / 13.1 / 48.4 | 7.7 / 18.8 / 61.0 | 8.1 / 22.5 / 72.0 | 7.1 / 19.0 / 63.2 | 8.3 / 20.2 / 63.9 | 7.6 / 20.6 / 65.3 | 6.7 / 17.4 / 58.8 |
| Multi-similarity [28] | 12.0 / 28.8 / 81.6 | 5.1 / 14.6 / 63.9 | 3.8 / 10.9 / 52.7 | 8.1 / 20.2 / 71.0 | 6.9 / 17.8 / 64.0 | 7.1 / 18.1 / 65.6 | 8.0 / 20.1 / 67.3 | 7.4 / 19.2 / 68.2 | 6.6 / 17.6 / 63.6 | 9.5 / 22.9 / 73.8 | 6.6 / 16.4 / 58.1 | 7.4 / 18.8 / 66.3 |
| Ours | 12.7 / 30.7 / 84.6 | 5.1 / 14.9 / 67.9 | 4.5 / 12.6 / 56.8 | 8.5 / 21.6 / 74.2 | 7.4 / 19.1 / 66.8 | 7.4 / 19.3 / 68.4 | 8.5 / 21.5 / 71.0 | 8.3 / 21.7 / 73.9 | 7.4 / 19.7 / 68.8 | 9.6 / 23.8 / 76.1 | 7.6 / 19.0 / 65.7 | 7.9 / 20.4 / 70.4 |

Table 2: Top-1 localization accuracy on RobotCar Seasons (top) and CMU Seasons (bottom).

loss [27], lazy triplet and quadruplet loss [30], and triplet loss with Huber distance [31]). We also compare to log-ratio loss [41] which, similarly to our method, avoids binary class labels. Finally, we compare to multi-similarity loss [28], which has partially inspired our method and is the current state of the art. In addition to these seven baselines trained by us on Oxford RobotCar, we also report results obtained with the checkpoints provided by [16] (off-the-shelf and triplet trained on Pittsburgh). Table 1 and the top half of Figure 4 show the baseline and state of the art comparison obtained with our evaluation setup on Oxford RobotCar and Pittsburgh, while Table 2 shows the results for RobotCar Seasons and CMU Seasons from the online benchmark. Our method performs best on all four test sets. In addition to the influence functions in (4) we also test linear and tanh functions as well as a simple sum alternative $\sum_{a \in \mathcal{A}} s_a^- - \sum_{a \in \mathcal{A}} s_a^+$ to (5). The results in the bottom five plots of Figure 4 indicate that a combination of (5) and (4) performs best. A set of qualitative results obtained by the proposed method and the compared ones is shown in Figure 3.

## 6    Conclusion

In this paper, we studied the problem of learning image features for retrieval-based visual localization and realized that there exists no theoretical framework that formalizes the process of learning such features. We showed why the existing ad-hoc based approaches are not suitable for the localization task. We introduced a formal problem definition and its formulation as a multi-objective optimization, and established the relationships between them. Our formulation allowed us to derive variants of loss functions, which were then used to train deep convolutional neural networks. We highlight one of those variants, for which our tests on four large-scale datasets (including one dataset created by ourselves) demonstrated a significant superiority over the state-of-the-art methods. We believe that our work opens up a new direction for learning image features targeted explicitly for localization. Our dataset and source code is made publicly available at `https://github.com/janinethoma/soft_contrastive_learning`.

## Broader Impact

This paper addresses the topic of retrieval-based visual localization. We present a formal problem statement and derive improved loss function variations for feature learning. While we are making our source code and data publicly available, we do not consider them a finished product. Any potential benefits and disadvantages depend on how people choose to use our method and how they handle failure cases. E.g. one obvious application of retrieval-based visual localization—pedestrian navigation—clearly benefits its users: they find their destination more quickly. A negligently or maliciously implemented navigation solution, however, could also lead people astray. Similarly, suppose our method is used in conjunction with a structure-based method for autonomous driving or robot navigation. In that case, it mostly leads to positive outcomes (e.g. increased safety and efficiency) as long as failure cases are adequately handled, and robots are not used for unethical tasks. To summarize, our method by itself does not have any direct negative ethical or social consequences. If it is integrated into a product, it is the quality and intention of the product that determine the broader impact.

The images used for training our networks were taken in Oxford. It is, therefore, to be expected that—if someone chooses to use our pre-trained models—they will work best in Oxford or similar places. The underlying theoretical formulation and loss functions, however, are entirely location-agnostic.

## Acknowledgments and Disclosure of Funding

This research has received funding from the EU Horizon 2020 research and innovation programme under grant agreement No. 820434.

## Footnotes

[1]Images captured with an intention to localize with a wide view of the surrounding.

[2]Images from different locations can observe the same place/landmark.

[3]Although our optimization seeks order preservation, our solution does not guarantee the desired ordering.

[4]Note that [28] does not make use of the geometric distances.

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
