[Supplementary Material · soft_contrastive_supplementary.pdf]

# Soft Contrastive Learning for Visual Localization
# Supplementary Material

**Janine Thoma**[1]           **Danda Pani Paudel**[1]           **Luc Van Gool**[1,2]

[1] Computer Vision Lab
ETH Zurich, Switzerland

[2] VISICS, ESAT/PSI
KU Leuven, Belgium

## 1   Background and Proofs

**Proposition 1.1 (Margin ambiguity, Proposition 3.3 in the paper)** *For a continuous mapping function $\phi_\theta : \mathcal{I} \to f \in \mathbb{R}^d$ and smooth manifold of features $\mathcal{F}$, when threshold $\tau_1 \to \tau_2$, Problem 3.2 is feasible only if $\alpha \to 0$.*

**Proof** When threshold $\tau_1 \to \tau_2$, two nearby images get assigned to the positive and negative classes respectively. For a continuous mapping function $\phi$ and a smooth manifold $\mathcal{F}$, this leads to $\min_{\mathcal{D} \in \mathcal{T}} \|n - p\| \to 0$. Let arguments $p^*$ and $n^*$ correspond to the optimal solution of $\min_{\mathcal{D} \in \mathcal{T}} \|n - p\|$. These arguments satisfy,

$$\|p^* - a\| \le \max_{p \in \mathcal{P}} \|p - a\| \quad \text{and} \quad \min_{n \in \mathcal{N}} \|n - a\| \le \|n^* - a\|. \tag{1}$$

Since $\|p^* - n^*\| \to 0$ when $\tau_1 \to \tau_2$, (1) implies the fallowing,

$$\max_{p \in \mathcal{P}} \|p - a\| \to \min_{n \in \mathcal{N}} \|n - a\|. \tag{2}$$

When (2) (here) is used in the context of Problem 3.2, the constraint of (2) (in the paper) is satisfied only when $\alpha \to 0$. Therefore, the Problem 3.2 is feasible only if $\alpha \to 0$. ∎

**Definition 1.2 (Feasible set)** *Let the mapping function $\phi_\theta : \mathcal{I} \to f \in \mathbb{R}^d$ map images into features using parameters $\theta$, which are learned using a set of tuples $\mathcal{T}$. If the desired parameters $\theta$ must respect*

$$\mathbf{d}(x_i, x_a) \le \mathbf{d}(x_j, x_a), \implies \|f_i - a\| \le \|f_j - a\|, \forall \mathcal{D}_i, \mathcal{D}_j \in \mathcal{T}, \tag{3}$$

*for anchor $a \in \mathcal{A}$ and proximity measure $\mathbf{d}(.)$, the feasible set of $\theta$ is the set of all $\theta$ that satisfy (3).*

**Definition 1.3 (Pareto optimal front)** *The pareto optimal front of the multi-objective optimization problem, $min_\theta [s_1, \ldots, s_i, \ldots]^\top$ is a set of parameters $\theta$ which minimize some objective $s_i$ such that $s_i$ cannot be further minimized without hurting any other objective $s_j$.*

**Proposition 1.4 (Duality, Proposition 4.2 in the paper)** *The pareto optimal front of the multi-objective optimization problem, $min_\theta [s_1^+, -s_1^-, \ldots, s_i^+, -s_i^-, \ldots]^\top$ passes through the feasible solution set of the Problem 3.1.*

**Proof** Let $\hat{\theta}$ be a feasible solution of Problem 3.1. Any deviation $\hat{\theta} + \Delta\hat{\theta}$ that minimizes any $s_i^+$, without hurting other objectives (corresponding to $\hat{\theta}$), still lies in the feasible set, if $g^+(0) \ge 0$. Any further deviation that maximizes any $s_i^-$, without hurting other objectives, ensures the feasibility of the solution. Note that some solutions near to the pareto optimal front are pareto optimal, and others are not. As long as any feasible solution deviates while minimizing one objective and not hurting the others, it goes towards the optimal front. That means, every feasible solution can deviate to the pareto optimal front while still being feasible. Therefore, the pareto optimal front passes through the feasible set. ∎

Figure 1: Top-1 localization accuracy for a fixed localization threshold $d = 10m$ as a function of the feature dimension after PCA and whitening.

Figure 2: Top-1 localization accuracy as a function of the distance $l$ between two consecutive reference images for query images from four different conditions of the Oxford RobotCar dataset.

## 2   Additional Quantitative Results

Figure 3 in our paper shows the top-1 localization accuracy as a function of the distance threshold $d$ for correct retrieval with a fixed feature dimension of 256. In Figure 1 we report the top-1 localization accuracy for a fixed threshold $d = 10m$ with varying feature dimension. It can be observed that the feature dimension has a strong impact on localization. The models trained on Oxford RobotCar are less sensitive to changes in feature dimension than the off-the-shelf models, which simply use ImageNet weights.

Another factor that can influence the localization accuracy is the spacing between reference images. In Figure 2 we linearly subsample the reference sequence and report the top-1 localization accuracy as a function of the distance $l$ between two consecutive reference images. In Figure 2 $d$ is set to 10m and the feature dimension is 256. A spacing of $l = 0m$ means that all available samples in the reference sequence are utilized. The results in 2 show that models trained on Oxford RobotCar are robust against an increase in the distance between reference images.

## 3   Sensitivity to Hyper-Parameters and Influence Functions

Fig. 4 shows the top-1 mean localization accuracy over all four conditions on Oxford RobotCar as well as Pittsburgh as a function of the distance threshold $d$ using our loss (5) for different influence functions and number of near/far samples per tuple.

Figure 3: Color code for t-SNE plots for the test regions from Oxford RobotCar and Pittsburgh. Best viewed on screen.

Figure 4: Top-1 localization accuracy on Oxford RobotCar (mean over four conditions) and Pittsburgh as a function of the distance threshold $d$ using our loss (5) for different influence functions and number of near/far samples per tuple.

# 4 Feature Visualization

To better understand the influence of the training loss function, we visualize the features learned by the different models in our paper using t-SNE and Grad-CAM representations.

## 4.1 T-SNE

T-distributed Stochastic Neighbor Embedding (t-SNE) [7] is a method for visualizing high dimensional data. It defines two probability distributions, one for pairs of points in the original high-dimensional feature space and one for pairs of points in the low dimensional embedding. Pairs of similar points are given a high probability and dissimilar pairs a low probability. The points in the low dimensional map are then found by minimizing the Kullback-Leibler divergence between the two distributions. This means that points that are located close in the embedding are—with a high probability—also close in the original feature space, while points with high distance in the embedding are most likely also dissimilar in feature space.

We use color to encode the original locations. The color codes are shown in Figure 3. Figures 5 and 6 show the t-SNE visualizations for the Oxford RobotCar and Pittsburgh test regions. Ideally, features that are close in geometric space should also be close in feature space. This means that the t-SNE plot of a good feature should cluster similar colors together. Looking at Figures 5 and 6 reveals that this is most prominently the case for models that also have a high localization accuracy according to the quantitative evaluation in Table 1 and Table 2 of our paper, such as our method.

## 4.2 Grad-CAM

We use Grad-CAM [8] to visualize which regions of a query image and the corresponding retrieved reference image contribute most to the match between query and reference image. In other words, we take the gradient of the negative squared feature distance between the query image and its top-1 retrieved reference image flowing into the last fully convolutional layer (5_3) to get a coarse heat-map of which regions are most important for the matching decision. Some selected results are shown in Figures 7, 8, 9, 10, and 11. We also provide a movie (grad_cam.mp4) with a comprehensive collection of Grad-CAM visualizations for our method compared to triplet trained on Oxford and off-the-shelf (initialization with ImageNet without localization training). The query images in the movie are linearly down-sampled to obtain a movie that fits the supplementary size requirements. The color of the reported distance between query and reference image indicates whether an image was localized within the specified threshold (10m).

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

Figure 5: T-SNE visualization of the query image feature distribution for different models. The colors correspond to different locations as shown in Figure 3. Continued in Figure 6. Best viewed on screen.

Figure 6: T-SNE visualization of the query image feature distribution for different models. The colors correspond to locations as shown in Figure 3. More results in Figure 5. Best viewed on screen.

Figure 7: Selected visual results. For each example, we show the query image, Grad-CAM on the query image, Grad-CAM on the retrieved image, retrieved image, and the distance between the retrieved image and the query image. Results obtained with our method are shown in the last row.

Figure 8: Selected visual results. For each example, we show the query image, Grad-CAM on the query image, Grad-CAM on the retrieved image, retrieved image, and the distance between the retrieved image and the query image. Results obtained with our method are shown in the last row.

Figure 9: Selected visual results. For each example, we show the query image, Grad-CAM on the query image, Grad-CAM on the retrieved image, retrieved image, and the distance between the retrieved image and the query image. Results obtained with our method are shown in the last row.

Figure 10: Selected visual results. For each example, we show the query image, Grad-CAM on the query image, Grad-CAM on the retrieved image, retrieved image, and the distance between the retrieved image and the query image. Results obtained with our method are shown in the last row.

Figure 11: Selected visual results. For each example, we show the query image, Grad-CAM on the query image, Grad-CAM on the retrieved image, retrieved image, and the distance between the retrieved image and the query image. Results obtained with our method are shown in the last row.