[Reviews · NeurIPS 2020]

Review 1

Summary and Contributions: This works tackles the problem of localization by image retrieval, specifically the training for such a problem. They argue that most of previous work focuses on artificial hard image split to positive and negative classes, and propose to do it in a soft manner. Namely, they propose a formal framework for learning localization features instead of place recognition features; formulate a novel loss for such a purpose; obtain state of the art performance.

Strengths: + I really like the problem formulation, it is done in a really principled way and it properly motivates proposed solution + Newly proposed loss utilizing soft assignments seems to be very useful in this particular task/area, and the community would benefit from it + Very good experimental comparison against state-of-the-art approaches, on many datasets, and under different image conditions + Nice ablation study on the choice of soft assignment function, it showcase the benefit of using the proposed "influence" functions and the final proposed loss

Weaknesses: - Last contribution, ie, dataset cleaning, is a very minor contribution and shouldnt be the part of major ones. Unless there is some specific novel methodology applied to clean the dataset, I dont think this contributions should be listed here. - Loss is inspired by [28], authors should be more clear on how much different it is, and clearly state which are the improvements. I dont think that is too clear at the moment. - Table 2 is very hard to read, some color coding of the 1st and 2nd best results would be good. Additionally, saying "Our method performs best on all four test sets." is not supported in the RobotCar Seasons datasets (table 2 upper part), as looking at the mean results the proposed method is not the best in 2 out of 3 metrics used. Nevertheless, overall, this method is state-of-the-art approach, the wording could be better, that is all. Minor: l262: multi-similarity loss [42] -> [28] l266: Table 1 -> 2

Correctness: Yes, seems correct.

Clarity: I find the paper to be clear, well written, and easy to follow.

Relation to Prior Work: Additional comments on difference to very related work [28] could help the reader, as stated in weaknesses.

Reproducibility: Yes

Additional Feedback: I like the work, and will propose to accept it. I think it benefits the area of place localization via image retrieval, gives a nice theoretical overview of the problem, and a novel solution that achieves state-of-the-art results. All issues I have are stated in weaknesses, and could be easily resolved at rebuttal time. I hope authors do that, in which case I will be overall happy to propose acceptance of the submission.


Review 2

Summary and Contributions: The paper tackles visual localization. It is standard to train image representations for localization via metric learning by defining positive and negative pairs based on some threshold. This paper proposes to instead of using hard positive/negative labels use a softened labels, e.g. nearby images are 'very positive', far away 'very negative', medium distance images are '50-50'. These scores are then integrated into the loss of [28]. Extensive comparisons show good benefits of the proposed method.

Strengths: 1) It makes sense not to have hard labelling of positives vs negative pairs, so making this soft is nice. 2) Overall results look good compared to many baselines / SOTA methods.

Weaknesses: 1) The training and evaluation protocols are somewhat non-standard and it's not clear why this is. E.g. L.193 talks about removing outliers and under/over-exposed images for training, L.208 about ignoring some images for evaluation. So essentially we can't compare the given numbers to any published results in the literature, despise using existing publicly available datasets for train and eval. Why not just use a standard procedure? E.g. train on the same data as others train on RobotCar? Or train on Pittsburgh? 2) A bit of a smaller point: It's not clear how fragile is the proposed method. For example, authors show figure 3 with tanh instead of sigmoid and this seems to significantly degrade results. It's a bit surprising as tanh and sigmoid are just scaled and shifted versions of each other. But I think sigmoid makes most intuitive sense to use so this might not be such an issue.

Correctness: All looks fine although I didn't check the proofs as they look intuitively correct (especially Proposition 3.3 which I think is fairly trivial)

Clarity: I think the paper is overly mathematical for what is actually proposed. It's simple (and that's not a negative): 1) instead of hard labels use soft labels, and you can use any function g+ and g- to define these soft labels depending on world distance, 2) insert this into loss of [28]. I don't think we need all this maths and propositions and proofs to present this simple idea, it makes it a lot harder to follow than it should be. For example, L.135 talks about positiveness and negativeness and I find these terms very confusing - assuming g+ is box or sigmoid, the ideal network should yield f_i=a for x_i=x_a so the 'positiveness' score should actually be low (0) for the most positive example. Then it should be larger than 0 for some 'less positive' but near-enough examples (e.g. distance=5). Finally it should again be 0 for very far away points. I think it would make a lot more sense in terms of exposition to just state that g+ and g- are essentially the soft labels and build from there.

Relation to Prior Work: Yes

Reproducibility: Yes

Additional Feedback: Results: - It's not quite true that L.267 "Our method performs best on all four test sets" - RobotCar Seasons Mean seems to show that Triplet works better for 2 out of 3 thresholds (and multiple other methods as well). - Table 2: a) it's way too hard to read on printed paper, b) please put best numbers in bold here as well as for table 1. - Table 1: it seems 'mean' is averaged over 4 subsets of RobotCar and Pittsburgh which is strange, it would be better to have 'RobotCar Mean' instead. Intro / related work / .. - I think there's a bit too much emphasis on 'arbitrarily chosen thresholds' that previous methods use. Are they really so arbitrary? In your production system you might want to define what success is, e.g. correct localization= up to 10m error (e.g. that's anyway how all evaluations are done!), so why is it arbitrary to use the 10m threshold at training to exactly optimize for what we want? Making training exactly aligned with evaluation sounds perfectly reasonable to me. Also for example "Visual Place Recognition with Repetitive Structures" Torii et al. states that the noise in GPS/StreetView is 7-15 m so a commonly used 25m threshold is not so arbitrarily chosen.. But I do agree that it's too harsh to say 9.99m is positive and 1.01m is negative so the proposed idea makes sense. - I'm not sure about the distinction in L.71 between (ii) and (iii). Why is e.g. [16] trained for place recognition and the proposed method is trained for localization? They both train with metric learning based on GPS distances. In fact, as explained in the paper, the hard labels with margin loss can be seen as a special case of what is proposed here, which should mean they fall into the same class of methods. --- Post rebuttal: I didn't need convincing, I think it is a solid paper. I think points in my review are still valid, e.g. I think the exposition is overly mathematical to the point that it hurts understanding, but it also comes down to personal preferences.


Review 3

Summary and Contributions: The paper is dealing with image-based localization handled as instance search. The goal is to learn good image embeddings that work well for search. Typically, the locations of the training data are used to make discrete groups of positive and negative images. The authors depart from that. They exploit the continuous nature of the training data coordinates and propose a continuous measure for positiveness and negativeness.

Strengths: - This is a novel approach, that is simple and easy to understand. - The authors fix a drawback of previous approaches. It is natural to exploit the continuity of the training labels. - The method is technically sound and comes with theoretical justifications. Other than that, it is intuitively understandable. - There is an extensive experimental validation comparing with prior losses. All are evaluated by the authors under the same conditions. The proposes method achieves top results. -

Weaknesses: - Many images that are nearby to the anchor will not share the same viewpoint. Therefore, they are not visually positives. This is handled in [16] by disambiguate using the image similarity according to the current descriptor. I am missing the discussion of this case in the paper. Can the authors comment? Is the approach of [16] not needed in their method? Can it still be complementary? - The paper does not include any variations or ablations of the proposed approach, but only restricts to comparisons with other approaches. I am missing some understanding of what are the key ingredients and what is the sensitivity of the method. For instance: -- it is said "Although any choice of strictly increasing and decreasing functions are eligible for influence functions, a careful choice is necessary for the desired outcome in practice.". How much does one have to tune the shape of the function? If it is very sensitive to that, then, tuning this instead of the distance thresholds in other approaches might even be less convenient. -- The number of nearby and far examples are kept fixed. What is the sensitivity to that? How many tuples does one batch include? Does it make sense to consider examples of all other tuples as far examples for the current tuple?

Correctness: I did not spot any flaws, but did not check all the derivations and proofs thoroughly.

Clarity: The paper is a pleasure to read.

Relation to Prior Work: The method is well positioned wrt prior work, and well motivated using the drawbacks of previous methods.

Reproducibility: Yes

Additional Feedback: questions to the authors are listed in weaknesses. --- After reading the other reviews and the rebuttal, I still believe the paper is worth to be published. The authors should include the experimental results of Figure 1 from the rebuttal (sensitivity of the approach to hyper-parameters and influence functions) in the final supplementary material, but in a more readable way.


Review 4

Summary and Contributions: This paper tackles the problem of visual geo-localization using metric learning. Given a set of images, features are learned so as to respect the ordering between their respective geographic distance. This is usually done using classical metric learning loss functions (contrastive, triplet, etc). In the paper, it is argued that defining a set of positive and negative images for a given query based on their geographic distance leads to an ill-posed problem. As such, they propose to relax the hard assignment label. They cast this into a multiple objective problem where all retrievable images have a positive and a negative score, weighted by monotonic functions of their geographic distance to the query. The multi-objective problem is reduced using a sum-log-exp aggregation. Experiments are performed on visual localization datasets.

Strengths: The formulation using a multi-objective problem is really nice and leads to a different interpretation of the metric learning problem. The paper is well written and well structured.

Weaknesses: Although the multiple objective is very nice, when aggregating using the sum-log-exp, it boils down to something very similar to multi-similarity. A clear analysis of the difference with MS should have been done. Also, what is the point of Proposition 3.3? Why would the case tau_1 -> tau_2 matter? Given that the training set is always finite in the studied problem, there will always be some small margin between tau_1 and tau_2. Or you want 2 images at exactly the same distance to be in different classes? Section 3.2 seems really off to motivate the choice of soft-assignement. Similarly, l.131 complains that the order is not preserved. This is not a problem of hard assignment but a problem of having reduced an ordinal regression problem into a binary classification problem. Learning to rank exists with methods able to preserve the ordering, which by the way the proposed method does not.

Correctness: The formal developments are correct, although some conclusions are deceptive (like the fact that hard assignment does not preserve ordering). Experiments are correct.

Clarity: The paper is well written, although the long string of references hinder readability. It is not enjoyable to have to skip to the reference section to discover what papers correspond to the ten numbers in brackets at the end of every other sentence Also, since the proofs are so small, they could have fitted the main paper.

Relation to Prior Work: The related work is sort of small in term of development. In particular, [31] is the paper that most likely answers the problems presented in later sections. It should have been more thoroughly explained and how the current work solves a problem that [31] does not. Similarly, as the method looks very similar to MS, a more detailed comparison should have been done.

Reproducibility: Yes

Additional Feedback: There are 2 things that to my mind would greatly improve the paper: 1. Better use the setup presented in the beginning of section 4, or remove it. As it is in the current paper, it is difficult to see the motivation for a multi-objective problem and how it solves the problems raised in section 3.2. 2. Make a better comparison with both [31] and MS since these are the 2 most related paper. What precise problems do they both have and how the proposed weighting scheme solves them. Final comments: I would like to thank the authors for providing a strong rebuttal, which together with the discussion with other reviewers made me update my review towards acceptance. I nonetheless still feel that the theoretical section is misleading in the sense that the proposed methods does not offer more guarantees than the literature. In particular, the proposed method does not guarantee that the solution is in the feasible set of problem 3.2 or that the corresponding feasible set has a margin >0. Maybe this could be clarified in the final version. The relation with multi-similarity is also very important and I hope to see it in the final version.

[Author Response · NeurIPS 2020]



Figure 1: Top-1 localization accuracy on Oxford RobotCar (mean over four conditions) and Pittsburgh as a function of the distance threshold $d$ using our loss (5) for different influence functions and number of near/far samples per tuple.

We thank all reviewers and ACs. We appreciate the many positive and constructive comments. Here, we first address
common concerns followed by reviewer-specific answers. We will update our paper and supp. material accordingly.

**Sensitivity (R2,R3).** Our results for tanh and (4) differ because (4) uses slope $\gamma$ and offset $\tau$ while for tanh we use
$y_n = \tanh \frac{x}{\zeta}, y_p = 1 - y_n$. Thank you for pointing out this missing detail. Following your suggestion, Fig.1 shows
that our method is not very sensitive with respect to the influence function and the number of near/far samples per tuple.

**Difference to MS [28] (R1,R4).** In contrast to our method, Multi-Similarity (MS) loss [28] requires hard class
assignments. Unfortunately, [28] was lost from the citations in L80, which may have made this unclear. The sum-log-
exp in (5) and [28] is a commonly used loss function. Our inspiration to use such a loss is based on [28] being a strong
baseline. In [28], the feature similarity kernels of positives and negatives are summed up in separate terms. To avoid
such a separation (which our paper argues to be unnatural for the task of localization), our loss weighs each feature
similarity with an influence function based on the geometric distance between images, resulting in our positiveness
and negativeness scores described in equation (3), which are then used in (5). To summarize, our loss uses geometric
distance weighting while [28] requires class labels, making our loss a more natural choice for the task of localization.

**R1.** Dataset contribution: We will move this from the main list of contributions to the experiments section. Still, we
believe that our curated dataset is valuable for the community and will therefore make it publicly available along with
the source code. Tab.2: We will add color coding, improve the wording of L267f, and correct the erroneous references.

**R2.** Mathematical complexity: The proposed method is indeed easily comprehensible, even without the mathematical
details. We will guide readers, who are only interested in a general understanding, to skip mathematical details.
Additionally, we will add an intuitive explanation similar to yours. Nevertheless, we believe that the multi-objective
formulation is important for an in-depth understanding of our interpretations. Evaluation: Tab.2 uses public evaluation
benchmarks (i.e. CMU Seasons and Robotcar Seasons). Therefore, these results can be directly compared to the
literature. Please note that "RobotCar Seasons" and "Oxford RobotCar" are different (Fig.2 in our paper). Only the
much larger latter is cleaned by us. Tables and L267: Please, see R1. Tab.1: The mean of Oxford RobotCar will
be added. Arbitrary thresholds: Thresholds based on system requirements are indeed not arbitrary. We will put this
into perspective. Localization vs. place recognition: The distinction between (ii) and (iii) on L71 is based on the
experimental setups and the underlying theoretical implications. Also, [16] refers to itself as a place recognition method.

**R3.** Viewpoint: We only consider positives with less than $30°$ yaw difference to the anchor (L232). This allows us to
learn from difficult positives (e.g. night), which are dropped by [16] whenever a tuple has more than one true positive.

**R4.** Prop.3.3: To avoid discarding images between $\tau_1$ and $\tau_2$ (L122), one may choose $\tau_1 \rightarrow \tau_2$. Prop.3.3 shows that
this choice contradicts with $\alpha > \epsilon$ in Prob.3.2. Even if there is always a small gap between $\tau_1$ and $\tau_2$ in finite datasets,
$\alpha$ may need to become very small for feasibility, which causes the notion of contrastiveness to disappear in Prob.3.2.
Sec.3.2 as motivation: The clarification of Prop.3.3 should resolves this issue. L131: The order is not preserved when
$\alpha = 0$. Ordinal regression vs. binary classification: We agree. The problem of binary classification directly relates to
the hard assignment of images into positive/negative (binary) classes. Therefore, our claim aligns with your analysis.
Learning to rank/order preservation: We are not aware of any learning method that *ensures* order preservation for the
problem at hand. Most methods attempt to maintain ranking, including ours (L146). Our Prop.4.2 sheds light on
this. Note that every feasible solution of Prob.1 is guaranteed to preserve the order. On the other hand, minimizing
(5) or similar, in an attempt to minimize the multi-objective function, provides a solution from the pareto optimal
front. Since the pareto optimal front passes through the feasible set of Prob.1 (Prop.4.2), it is possible that for some
settings of (5), the obtained solution also preserves the desired order (due to being feasible for Prob.1). Please, see
L144. Sec.4 in relation to Sec.3.2: We show in Prop.4.2 that our multi-objective formulation may lead to a feasible,
order preserving solution of Prob.1. The formulation of Prob.3.2, on the other hand, is problematic in the context of
visual localization as shown in Prop.3.3. We will add these high-level connections, to further improve the clarity of our
paper. Comparison to [31]: Unlike our method, [31] does not use soft assignments (L77). Their negative visual loss
uses hard assignments. Their visual-geometric loss uses geometric distances but is only applied to positives (L61).

[Meta-Review · NeurIPS 2020]

The initial scores for this paper were: 5: Marginally below the acceptance threshold. 7: A good submission; accept. 7: A good submission; accept. 7: A good submission; accept. Reviewers are overall positive but raise issues regarding clarity of explanation, missing discussion, some missing ablations, and some missing analysis of the results. The main issue raised by R4 (5) is regarding the need and the justification of the theoretical part of the paper. The authors provide a rebuttal. In the post-rebuttal discussion, R4 raises their score from 5 to 7 but their issue regarding the theoretical section of the paper partly remains. The final scores of the paper are 7, 7, 7, 7. The reviewers provide final comments after seeing the rebuttal in their reviews. The AC is convinced by the positive arguments of the reviewers and recommends Accept. The authors are strongly encouraged to take into account all reviewers' feedback (and especially the remaining comments mentioned by R4) when preparing the final version.